# Impact of delayed response on wearable cognitive assistance

**Manuel Olguín Muñoz**[1]*, **Roberta Klatzky**[2,3], **Junjue Wang**[4], **Padmanabhan Pillai**[5], **Mahadev Satyanarayanan**[4], **James Gross**[1]

**1** Division of Information Science and Engineering, School of Electrical Engineering and Computer Science, KTH Royal Institute of Technology, Stockholm, Sweden, **2** Department of Psychology, Carnegie Mellon University, Pittsburgh, PA, United States of America, **3** Human-Computer Interaction Institute, Carnegie Mellon University, Pittsburgh, PA, United States of America, **4** School of Computer Science, Carnegie Mellon University, Pittsburgh, PA, United States of America, **5** Intel Labs, Pittsburgh, PA, United States of America

* molguin@kth.se

**Data Availability Statement:** The full dataset for the research has been made publicly available on the Zenodo platform; URL: https://zenodo.org/record/4494912; DOI: 10.5281/zenodo.4494912.

## Abstract

Wearable cognitive assistants (WCA) are anticipated to become a widely-used application class, in conjunction with emerging network infrastructures like 5G that incorporate edge computing capabilities. While prototypical studies of such applications exist today, the relationship between infrastructure service provisioning and its implication for WCA usability is largely unexplored despite the relevance that these applications have for future networks. This paper presents an experimental study assessing how WCA users react to varying end-to-end delays induced by the application pipeline or infrastructure. Participants interacted directly with an instrumented task-guidance WCA as delays were introduced into the system in a controllable fashion. System and task state were tracked in real time, and biometric data from wearable sensors on the participants were recorded. Our results show that periods of extended system delay cause users to correspondingly (and substantially) slow down in their guided task execution, an effect that persists for a time after the system returns to a more responsive state. Furthermore, the slow-down in task execution is correlated with a personality trait, neuroticism, associated with intolerance for time delays. We show that our results implicate impaired cognitive planning, as contrasted with resource depletion or emotional arousal, as the reason for slowed user task executions under system delay. The findings have several implications for the design and operation of WCA applications as well as computational and communication infrastructure, and additionally for the development of performance analysis tools for WCA.

## 1 Introduction

Wearable Cognitive Assistants, WCA for short, have recently started to garner attention from the research community [1, 2]. They represent a novel category of highly interactive and context-sensitive augmented reality applications, that aim to amplify human cognition in both day-to-day tasks and professional settings. Their mode of operation is analogous to how GPS navigation systems guide drivers, by seamlessly providing relevant instructions and feedback

**Funding:** Funding 1: - Awarded to: James R. Gross - Grant number: ITM17-0246 (ExPECA) - Funder: Swedish Foundation for Strategic Research (SSF) - Funder Website: https://strategiska.se/ - Disclaimer: The funders had no role in study design, data collection and analysis, decision to publish, or preparation of the manuscript. Funding 2: - Awarded to: Mahadev Satyanarayanan - Grant number: CNS-1518865 - Funder: United States National Science Foundation (NSF) - Funder Website: https://www.nsf.gov/ - Disclaimer: The funders had no role in study design, data collection and analysis, decision to publish, or preparation of the manuscript.

**Competing interests:** The authors have declared that no competing interests exist.

relating to the current task at hand. Note that this implies seamless interaction with the context of the user—at no moment does the user need to trigger an update explicitly, as the application is constantly tracking the state of the target task. An example is an IKEA Assistant [3] which monitors the assembly of a piece of furniture in real time, providing timely, step-by-step feedback to guide the user toward completion.

WCA systems were originally inspired by assistive use cases for people suffering from some form of cognitive decline, either through aging or because of traumatic brain injuries [1, 4]. More recently, they have been applied to a broader range of use cases, including step-by-step guidance on complex assembly tasks [5]. Non-wearable augmented reality and cognitive assistance systems have already been proven to be valuable tools in the industrial workplace [6, 7]. Detethering this assistance from its current fixed location will surely make it available to many more fields.

Based on these use cases, we identify three main requirements for WCA:

1. WCA systems should be available whenever the user requires them, without being tethered to a particular physical location. Assistants need to be pervasive and mobile.

2. Interaction with the system should be immersive and seamless, i.e. the assistant should be able to analyze the current context and automatically provide relevant feedback without explicit commands from the user. In this sense, WCA is expected to operate much like a human assistant would, by observing the performance of the user and offering guidance proactively.

3. Feedback should be "quick", relative to the task at hand. This requirement is further strengthened by the previously mentioned "seamless interaction" characteristic of these systems. This means that users will have expectations of constant, immediate feedback as they progress through the task. In the case of a step-by-step task like IKEA, delayed feedback might simply confuse or distract the user. However, in a highly interactive task like a *Ping-Pong assistant* [2, 8] late guidance is at best a nuisance and at worst a severe handicap.

Item 1 implies use of lightweight and low-power devices, preferably a wearable device that frees both hands for work. Item 2, on the other hand, suggests a level of context sensitivity and proactivity that can only be provided by real-time analysis of sensor inputs such as video and audio feeds. The compute-intensive processing suggested by Item 2 cannot be met by the lightweight wearable devices suggested by Item 1. Only by offloading computation from a wearable device to cloud-based or edge-based infrastructure can this circle be squared. However, offloading implies an extended end-to-end pipeline with many potential sources of queueing, transmission, and processing delays. Item 3 therefore emerges as a key concern, requiring deep understanding of the impact of end-to-end delays on WCA users.

Item 3 forms the base motivation for the research presented in this paper. We still have a very limited understanding of how humans react to delays in these systems—specifically, how changes in system responsiveness impact users. *System responsiveness* here denotes a qualitative scale ranging from *high* (that is, not subject to delay or subject to negligible delay with respect to human perception) to *low* (i.e. subject to considerable delay). Characterizing the relationships between system responsiveness and user behavior and quality of experience is of paramount importance for the design and evaluation of these applications. It is generally acknowledged, for instance, that a system going from a state of high responsiveness to one of low responsiveness can cause a drop in quality of experience [9]. Furthermore, this could cause users to modify the temporal paramaters of their behavior when interacting with the system, generating a sort of *feedback loop* between user and system. A clear understanding of

these relationships would allow, for instance, for the development of strategies for load balancing and optimization for large-scale deployment of WCA.

This paper builds upon preliminary work in the field of time perception and delay characterization of WCA. We expand upon the findings of Ha et al. [1], who identified the need for low-latency offloading in WCA, and of Chen et al. [5], who outlined the bounds for "noticeable" and "unbearable" latencies in these systems. While these bounds present a general understanding of when it is likely that a user will stop using an application, they do not provide any insights as to what happens *before* that—i.e. how human behavior changes with system responsiveness. We aim to tackle this gap in knowledge through the characterization of human responses to delays in the application pipeline, using latencies in the range defined by the previously established bounds. This is an important step toward a more systematic understanding of human behavior in this domain.

We present in this paper the design and elaboration of an experimental WCA test-bed. This test-bed was subsequently employed in a study in which undergraduate students of diverse fields of study, aged between 18 and 25 years old, were asked to interact with and follow the instructions given to them by a cognitive assistant. Unbeknownst to the participants, we altered the responsiveness of the system in real-time and recorded the resulting behavioral and physiological reactions. The participants wore an array of biometric sensors measuring physiological responses that have proven useful in assessing cognitive workload during human-computer interaction [10, 11] such as heart rate and EEG.

Through this experimental set-up, we intended to answer four core research questions relating to human responses to decreased application responsiveness.

- *Do subjects change the temporal profile of their actions in relation to system latency?*
  In line with previous research in this area, we expected subjects to change their temporal profiles as system responsiveness decreased. The extent or form of these changes were however unknown. We also hypothesized that large enough drops in responsiveness could lead to complete abandonment of the task by subjects.
  Our results show an emergent pacing effect on user actions as system responsiveness is reduced. While it would seem self-evident that users take longer to complete a task while using a system with low responsiveness—as they have to wait longer for new instructions— our study found that user slow-down represents a source of substantial additional delay. To be more precise, the data indicate that users slow down not only because they have to wait for the system to catch up, but that their reactions to new instructions is also delayed. Moreover, this effect scales with the decrease in responsiveness and remains for a while, even after system responsiveness improves.

- *Do subjects show signals of arousal in physiological responses to changes in system latency?*
  We hypothesized subjects would show signs of stress and frustration as system latency increased, due to the added annoyance of dealing with an unresponsive system.
  The results we present, however, seem to refute this hypothesis. We were not able to detect any significant effects on the physiological signals obtained from the biometric sensors as sytem responsiveness was altered.

- *Are responses to delay effects in subjects mediated by cognition and/or emotion?*
  In line with previous items, we expected delay effects in subjects to be mediated primarily by emotion. In particular, we expected emotional effects to be correlated with the strength of the added delay.
  The results point in a different direction though, indicating that reduced responsiveness in WCA systems leads to a disruption of participants' cognitive plan for the task and not to an

emotional response. This is evidenced by the previously discussed pacing effect and the lack of significant physiological responses.

- *Are these effects mediated by personality indicators in any way?*
  Finally, we hypothesized that the individual trait of *neuroticism* [12] would play a role in mediating these effects, as it has previously been connected to intolerance for time delay [13]. We also expected *focus* and *involvement* [14] to play roles in this.
  The results obtained agree with our hypothesis. We found significant effects of neuroticism on the responses exhibited by subjects, and all three traits were found to play a role through factor analysis.

We believe that these results provide concrete and relevant implications for WCA design, deployment, and optimization. One example is the behavioral slow-down, as it extends application runtime significantly, and thus has clear and direct implications for resource and power consumption. Another is the fact that the adverse effects of delay on users do not immediately subside as delay is diminished—this has potential consequences for resource allocation strategies. Moreover, in multi-user scenarios, the dependency of user slow-down effects on delay mean efficient resource allocation across applications potentially looks different from what could be considered "fair".

Our hope is that the results we provide might prove useful for the understanding and optimization of deployments of WCA. These results represent unexpected, valuable findings, which can be employed to model and understand how users interact with latency in applications and systems, and develop resource allocation and power optimization strategies. Additionally, we hope that the results we provide might pave the way for the improvement of performance evaluation tools such as our previous work in [15, 16]. Such systems would greatly benefit from this knowledge, as it would allow for the design and implementation of realistic models of human behavior, making highly accurate benchmarks a reality in the domain of WCA.

The structure of this paper is as follows. We describe the existing body of research around time perception and the effects of delay on human performance in Section 2. Section 3 presents the experimental design, measures, and specific protocol. Then, in Section 4 we detail the results of our experiment. Implications for further modeling of the effects of delay are presented in Section 5 before finally concluding the paper in Section 6.

## 2 Background and related work

Our overarching theoretical construct is the idea that people perform temporally sequential tasks in the framework of cognition and emotion. In particular, we view cognition as described by models such as ACT-R [17]; i.e. as a sequence of procedures performed by the subject. Additionally, we hypothesize that individual diferences between subjects moderate responses to delays in the feedback during the execution of a task.

Accordingly, in the following we review background work in the areas of (Section 2.1) time perception (Section 2.2) mechanisms relating human performance to delay.

### 2.1 Time perception in computing systems

The question of how people respond to delay in a computer system is grounded in how people perceive time. Time perception has been described as regulated by an attentional gate that, when opened, starts a cognitive pulse counter [18, 19]. More recent research indicates, however, that duration perception is highly malleable and the result of multiple timing mechanisms found in overlapping, flexible neural systems [20, 21]. The estimation of an event's

duration varies with context of various types (i) events subsequent to a long or short interval are contracted or extended, respectively [22] (ii) repeated events tend to be perceived as shorter than novel ones [23] arousal can expand durations [24].

Expectations play a critical role in time perception as well [18, 19]. It has been shown that people have a general tendency to be hypersensitive to delays in worse-than-expected states, and under-sensitive to meeting or exceeding expectations [25]. Accordingly, failures to meet expected fast response times tend to be experienced as highly negative, whereas fast latencies are not noticed. Violations of expectancy have a strong impact on the acceptability of computer systems. Users of a computer system anticipate the latency for events, for which the standards only become more stringent as systems improve in response time. In immersive systems like WCA, which aim to provide seamless interaction, delays are particularly noticeable.

It has long been recognized that slow system response times can undermine cognitive processing, slow the pace of users, and lead to stereotyped behavior and errors, as well as cause negative emotional consequences [9]. However, standards for what constitute tolerable delays have changed dramatically compared to three decades ago, when delays on the order of 10 s were deemed acceptable [26–28]. Today's user context, and WCA in particular, often demand response times orders of magnitude shorter.

For WCA the acceptable range for latencies was explored by Chen et al. [5], by constructing assistants for tasks with a range of time constraints, including step-by-step tasks and more interactive contexts like playing Ping-Pong against a human opponent. They then proposed a latency tolerance zone according to the task demands. For an essentially self-paced task like LEGO assembly, they found two key ranges of latency; unnoticeable, 0 to 0.6s; and impaired, 0.6 to 2.7s. Beyond that, users could begin to show the negative outcomes previously catalogued [9].

## 2.2 Mechanisms relating delay to human performance

While behavioral changes and negative interaction outcomes have been well documented in prior research on system delay, the specific mechanisms that mediate these outcomes are less well understood. These mechanisms could be cognitive or emotional in origin.

A first possible explanation comes from research on cognitive and motor planning. Delay may move users from relatively automatic to more attention-demanding processing. Cognitive and motor tasks are commonly described as a hierarchical system, progressing from high-level goals to the sequence of commands that accomplishes them. As competency in a task increases, execution of the hierarchy becomes increasingly automated. Automatization has been described from a computational perspective in Anderson's ACT-R model as the compiling of multiple productions into one [17]. Neural measurements indicate that with automaticity, control moves from frontal brain areas to more posterior ones [29, 30], and similar distinctions have been related to temporal processing [31–33].

Although activities guided by a WCA are not simple motor actions, immediate feedback after each of a series of repeated actions should promote development and automatic execution of a hierarchical plan. Delays, in contrast, would disrupt such a plan through the loss of automated control [33].

A second, alternative explanation of delay effects appeals to emotional systems rather than cognitive processes. As users of a system become emotionally aroused by delay, they may be subject to generalized arousal, causing decrements in performance [33].

Finally, a third potential explanation of delay effects is what has been called "ego depletion", the notion that expending effort on self-control eliminates resources needed for further effort [34, 35].

The various processing accounts of delay effects predict different outcomes, which we will consider in the context of the current data. If delay increases attentional demands on cognitive processes, responses should be slowed and errors expected, particularly on time-critical tasks. Generalized arousal triggered by emotional stress from delay should emerge in physiological measures, such as increased heart rate or skin conductivity. Arousal can also reduce movement smoothness or add erratic gestures [36]. Ego depletion has been found to produce premature responses culminating in error [35], or to lead to abandoning a task entirely [34].

Over-arching prescriptions for tolerable system response time have not tended to take into account individual differences in users with respect to salient variables like cognitive ability or personality. Relevant research can be found in studies of delay discounting, the tendency to devalue rewards for which one must wait. High discounting rates, indicative of waiting intolerance, have been associated with negative social and academic outcomes. Hirsh et al. [13] found that higher discounting was associated with extraversion among those with low cognitive function, whereas lower discounting was associated with emotional stability (low neuroticism) for people with high cognitive function. Among computer system users who tend to have relatively high cognitive ability (which presumably describes the present experimental population), this points to neuroticism as a personality factor that might modulate tolerance for waiting. Extraversion could also be a moderating factor among the broader target audience of WCA, which are intended for relatively inexperienced users of an application. These and other measures of individual variation were considered here.

## 3 Experimental design

The core elements of our experiment are shown in Fig 1:

- Subjects interact with a WCA while wearing an array of biometric sensors.

- The *responsiveness* of the application, i.e. the interval of time between an input being provided to the system and the associated output returned to the user, is manipulated in real time. The effects of these manipulations on the subjects are recorded and subsequently analyzed.

This study was conducted with the approval of the Carnegie Mellon University Institutional Review Board under record number STUDY2019_00000247. Written consent was obtained.

Subjects were recruited from a pool of undergraduate students at Carnegie Mellon University. Students enrolled in an introductory-level psychology course fulfill a research requirement as part of the plan of study. This requirement may be fulfilled either through the elaboration of a written essay or by volunteering as a participant in a small number of research experiments.

No particular exclusion criteria were applied, and as specified by our approved data-collection protocol, no gender- or sex-related statistics were collected. In total, 40 participants were recruited, all of them of college student age (18-25 years old).

### 3.1 The cognitive assistance application

We used a modified version of the LEGO Assistant application introduced by Chen et al. [2]. This application belongs to a category of WCAs designed to guide users through the execution of a sequential task. The assistant presents a series of instructions to the user in a semi-predetermined order while it monitors the progress in real-time. Whenever the application detects that the user has correctly performed an instruction, it provides a new one. In the case of

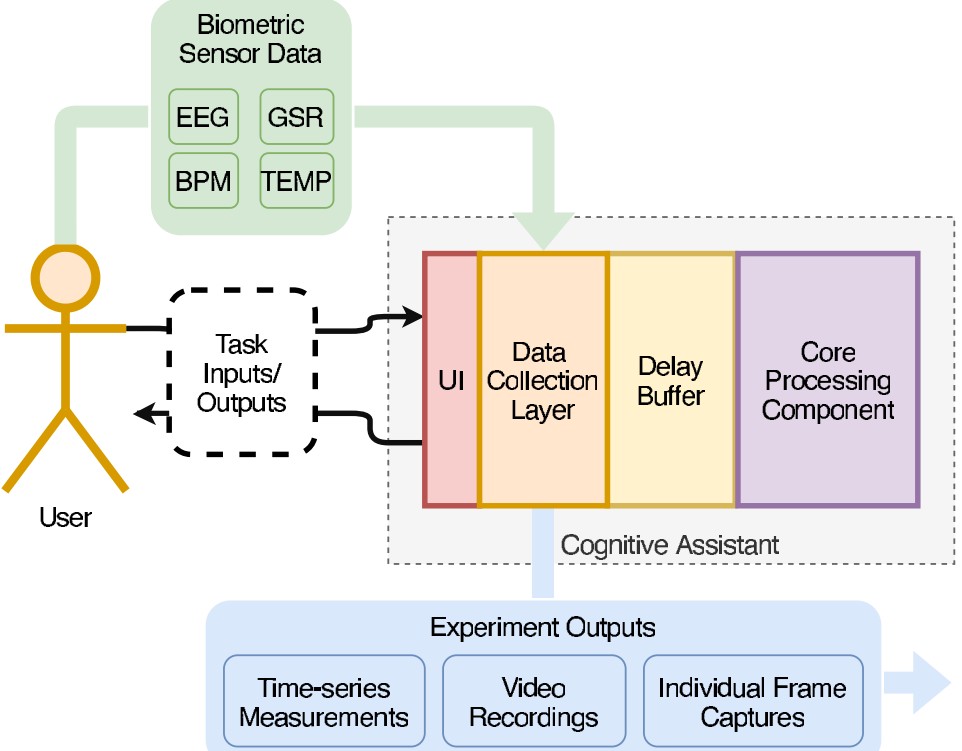

**Fig 1. Experimental test-bed.** Participants interact with the cognitive assistant through task-related inputs and outputs—in practice, these correspond to the video feed captured by the assistant and the instructions provided by it. The assistant itself has been instrumented with a data collection layer, which collects and processes experiment-related data such as biometric signals from the participants (these are merely processed here and do not form part of the inputs to the cognitive assistant as such, however), and a delay buffer, which introduces controlled delays in the transit of information from the core processing component.

an incorrect action by the user, a procedurally generated corrective instruction is provided instead.

In the following, we will provide some formal definitions relating to the WCA application. A *task* will be understood as a finite sequence of instructions to be performed in order. *Step* will refer to a specific action to be performed by the user, described by a single instruction; e.g. *put a green 1 × 3 brick on the board* (see Fig 2). A step starts with an instruction being given to

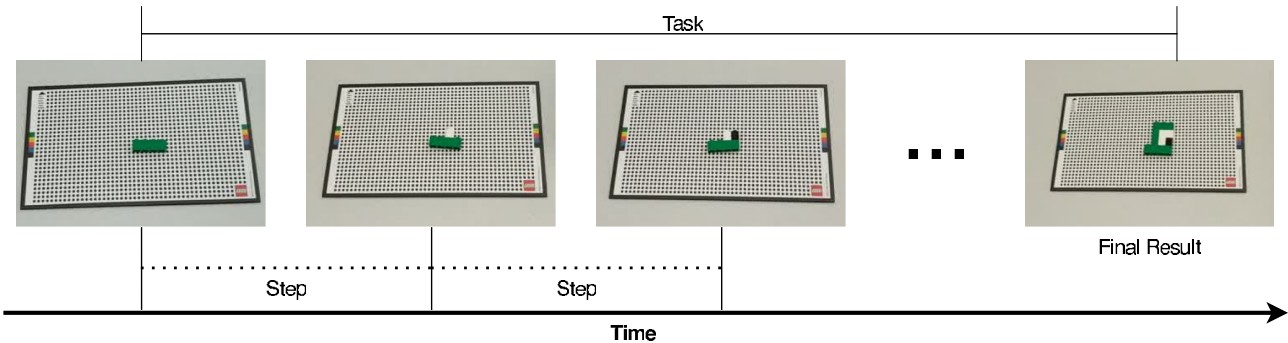

**Fig 2. Example of a cognitive assistance task and its component steps.**

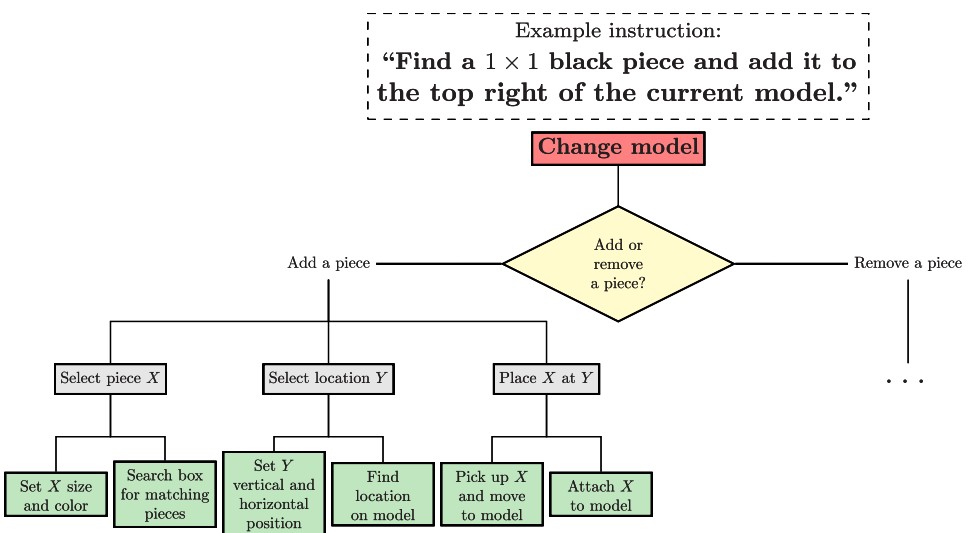

**Fig 3. Hierarchical cognitive structure of a step in the LEGO task.**

the user, and ends when the assistant detects that the user has finished performing it. At that point, the assistant either provides a new instruction to the user, or the task ends.

In the base LEGO Assistant, the task consists of the assembly of a LEGO model; each step requires the user to append a LEGO brick to the model at a specific location and orientation. The system monitors progress through a video feed and provides timely feedback in the form of visual and textual instructions to guide the user.

The LEGO assistant has features that make it a good target for assessment of delay effects in a cognitive assistant. In particular (1) it is easily understood by users and requires essentially no training (2) its step-by-step nature simplifies the isolation of the relevant experimental variables and the effects of the delay (3) each step has an intrinsic hierarchical cognitive structure (see Fig 3), affording multiple levels of cognitive control.

For the purposes of this study, the original client-server design of the LEGO Assistant was altered to be locally executable. This was done in order to eliminate the stochastic effects of jitter and latency from the network link, and greatly simplify instrumentation. Instructions were output in image and text form to a computer display situated on a table directly in front of the participants. Participants performed the instructions on the table, these actions being captured by a high-definition camera located on top of the display.

## 3.2 The experimental LEGO task

For the experimental LEGO task, a key modification was made to the structure of the steps. After each the processing of each input is completed, the result is withheld for a variable period of time until a specific target *delay* is reached, as illustrated in Fig 4a and 4b. The length of this delay is one of two independent variables we manipulated for the experiment. Seven levels of delay were used—no added delay (which we will also refer to as 0 s delay), 0.6, 1.125, 1.65, 2.175, 2.7, 3.0s—chosen based on the latency bounds found previously by Chen et al. [5]. Our selection of delays is centered around the range of delays where latency is noticeable to users, but the application remains in a "usable" state.

In order to study the effects of a delay applied across multiple steps we implement an experimental design component called a *block*, corresponding to a sequence of consecutive steps within a task subject to the same delay; see Fig 4c. The *length* of blocks is the second of the two

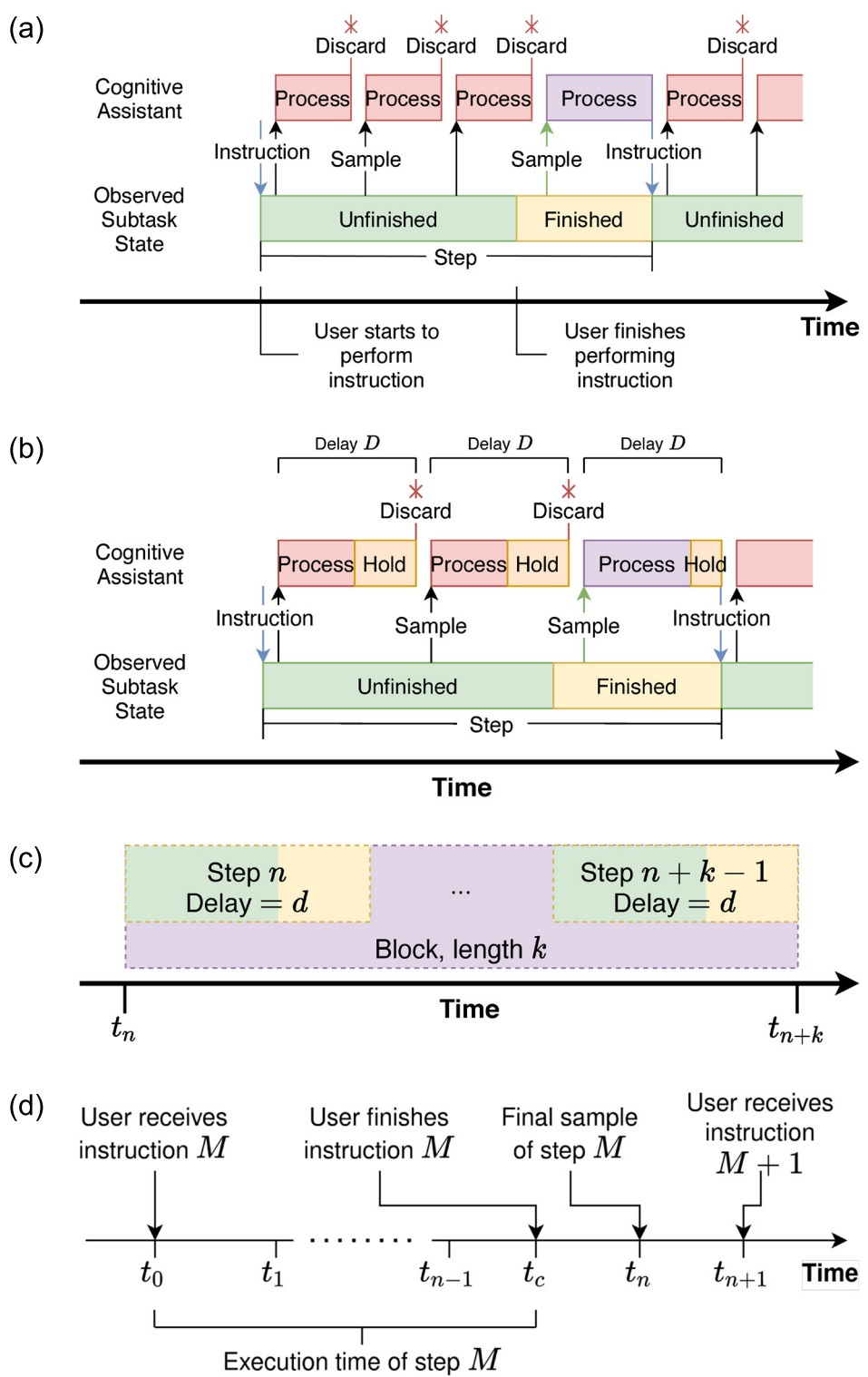

**Fig 4. Components of the cognitive assistance task.** (a) Structure of a step in a generic cognitive assistance application. The assistant provides an instruction to the user and continuously samples the step state; inputs captured while the step is unfinished are silently "discarded" (i.e. they do not cause the generation of a new instruction). Once the user finishes performing the step, the next sample *will* cause the generation of a new instruction. (b) In the experimental task, an additional variable segment of time is introduced immediately following the processing of the input frame in order to extend the perceived processing time of the input to a specific target delay. (c) Structure of a block in the experimental task. (d) Visualization of the execution time of a step.

independent variables manipulated in our experiment. We used values of 4, 8, 12 steps, chosen as representative of the number of steps in tasks in the base LEGO Assistant application. Additionally, we define the *duration* of a block, as the time elapsed between the start timestamp of the first step in the block and the end timestamp of the final step in the block.

To assign tasks to the participants, a pseudo-random permutation of the combinations of block length and delay was first generated, and a unique sequence of steps assigned to each of these combinations to create 21 unique blocks. A *Latin square*-type design was then used to reorder this initial permutation in order to generate a task for each participant. This ensures a counterbalance of the order of the blocks across participants and avoids systematic learning effects. Note that unlike the base LEGO assistant task, in which instructions led participants through the assembly of a specific model, the experimental LEGO task consisted of a sequence of instructions to either add or remove pieces with no evident goal.

Given the variability in the system latency to detect step completion, correcting step-completion time for system response time by a fixed amount would not enable sufficient precision. We therefore define the *execution time* empirically for an individual step as the total time between the user receiving the instruction for the step and the time instant of the sample capturing the completed step.

For an arbitrary step, we define the sequence $\{t_0, t_1, \ldots, t_n\}$ as the sequence of timestamps corresponding to the sampling instants during the step (see Fig 4d). $t_0$ corresponds to the instant when the instruction for the step is given to the user (and the first sample is taken), and $t_n$ to the timestamp of the last sample before a new instruction is given. If we define $t_c$ as the instant marking when the user presented the finished step to the system, it must follow that $t_{n-1} < t_c < t_n$. Due to the discrete sampling of the task state, there remains some imprecision in the estimate of execution time relative to $t_c$. However, this introduces no bias in the results, as we can assume that $t_c$ is uniformly distributed in the range $(t_{n-1}, t_n)$. We therefore calculate execution time for each step as $t_c - t_0, t_c \in U(t_{n-1}, t_n)$, which on average works out to an adjustment of the observed time by 1.5 times the mean sampling rate of the step.

## 3.3 Collected data

The collected data from the experiments fall into four categories: behavioral and personality indicators, *frame-to-frame* metrics, biometric data, and video recordings.

**3.3.1 Behavioral and personality indicators.** Before beginning the experimental procedure, participants were asked to fill out two questionnaires.

The first of these, the Big Five Inventory of Personality (*BFI* [12]), consists of 44 questions to be answered on a 5-point *Likert*-type scale, assessing the traits of *agreeableness: detached to compassionate* (8 questions), *conscientiousness: careless to organized* (9 questions), *extroversion: reserved to outgoing* (8 questions), *neuroticism: secure to sensitive* (8 questions), and *openness: cautious to curious* (9 questions). Of these, extroversion and neuroticism have been related to tolerance for delay [13].

The second survey, the Immersive Tendencies Questionnaire (*ITQ* [14]), comprises 29 questions, 28 of which were used for the study (one categorical question was disregarded), assessing the sub-scales of *involvement*, the tendency to become involved in activities; *focus*, the tendency to maintain focus on current activities, and *games*, the tendency to play games. These questions use a 7-point horizontal scale with opposing descriptors anchoring the ends. Participants were asked to mark the appropriate point in the scale, and these responses were converted to a numerical value between 1 and 7 for processing.

In post-processing, the obtained scores for both questionnaires were normalized to fall in the [0, 1] range for ease of interpretation. See Table 1 for their means and standard deviations.

**Table 1. Means and standard deviations of normalized questionnaire scores.**

| Questionnaire | Metric | Mean | SD |
|---|---|---:|---:|
| BFI | Agreeableness | 0.705 | 0.136 |
| | Conscientiousness | 0.562 | 0.149 |
| | Extroversion | 0.491 | 0.180 |
| | Neuroticism | 0.524 | 0.230 |
| | Openness | 0.677 | 0.152 |
| ITQ | Focus | 0.626 | 0.126 |
| | Games | 0.463 | 0.261 |
| | Involvement | 0.568 | 0.175 |
| | Total | 0.540 | 0.092 |

**3.3.2 Frame-to-frame metrics.** During the execution of the task, we logged every event occurring in the application pipeline. Each incoming frame (including frames discarded by the assistant), as well as its associated outputs, was logged at multiple points in the process along with associated metadata such as currently implemented delay as specified by the experimental design. This allowed us to extract metrics relating to the performance of the task, such as the time spent by participants on particular steps, any mistakes made, etc. In particular, it made possible the easy segmentation of the other time-series data we collected into our main unit of analysis, the aforementioned *block*.

**3.3.3 Biometric data.** The participants wore devices to acquire four physiological measures: (1) galvanic skin response (*GSR*) (2) accelerometer data from the dominant wrist (3) brain activity in the form of electroencephalography (*EEG*) (4) heart rate.

These metrics have been used as indicators of stress and cognitive load by an ample body of previous research [37–39]. GSR (also known as electrodermal activity) can be interpreted as an indicator of physiological arousal and has long been a widely used metric in studies seeking to characterize mental workload [37–42]. EEG on the other hand has previously been used to measure cognitive load in the context of human-computer interactions [11, 43, 44].

Wrist acceleration, GSR and heart rate data were obtained using the Empatica E4 [45] bio-sensing wristband. Accelerometer data was sampled at 32 hz, GSR was sampled at 4 hz, and instantaneous heart rate was calculated from a *blood volume pulse* (BVP) signal sampled at 64 hz. Participants were asked to wear the device for approximately 10–15 minutes before starting the experiment, in order to allow the sensors to reach a stable equilibrium and establish a baseline for the signals.

The E4 wristband was chosen due to its small, non-invasive, and wireless form factor (samples were streamed to the system over Bluetooth LE) and for the fact that its use in research has been experimentally validated in previous studies [46, 47]. The E4 also includes a skin temperature thermometer, however the measure was not used for the present study.

For the EEG data we employed the OpenBCI EEG Headband Kit [48] consisting of a number of dry electrodes fastened to a Velcro headband. It provides a quick and non-invasive way of obtaining EEG signals from participants. Electrodes were placed according to the *10–20 Electrode System* [49] on the *Fp1* and *Fp2* points, in order to capture brain activity in the frontal lobe, with ground and reference electrodes on the right and left earlobes, respectively. The signals were sampled at 200 hz and postprocessed to (1) remove frequencies outside the 0.1—40 hz range (2) smooth out noise (using the technique proposed by Agarwal et al. [50]).

**3.3.4 Video recordings.** During the task, participants were recorded by two separate cameras. One camera was angled downwards, towards the table, the LEGO board, and the

(a)

(b)

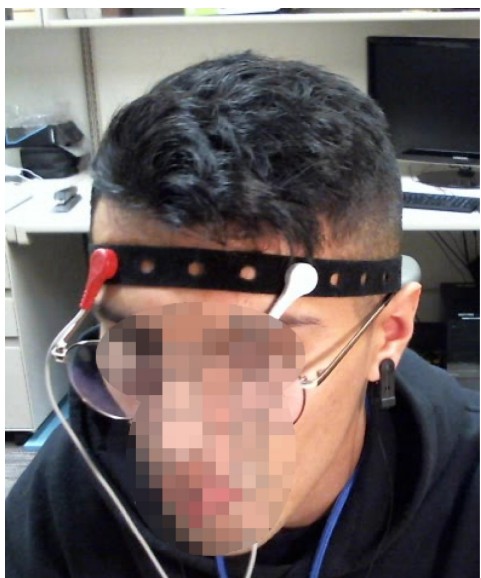
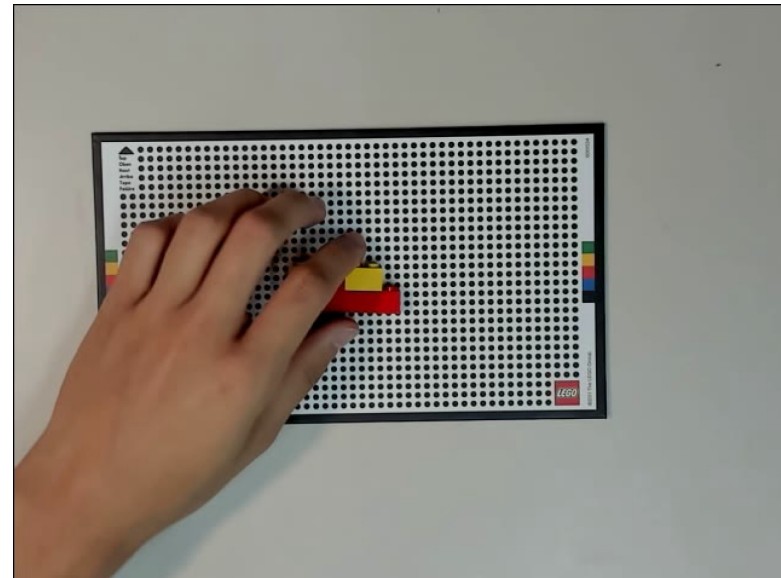

**Fig 5. Sample frames captured during the experiment. (a)** Frame from the face recording of a random participant, clearly showing the locations of the EEG electrodes. **(b)** Frame from the board recording of the same participant.

participant's hands. This camera was used to capture the necessary inputs for the LEGO assembly task as well as to record the actions performed by the user. The second camera was angled horizontally, towards the participant, in order to record their facial expressions during the execution of the task.

Both video feeds were captured at a rate of 24 FPS in parallel processes to ensure a constant rate of capture. Examples of the captured frames can be seen in Fig 5. The video feeds were not used for the present study, but may be utilized in future analysis.

## 4 Results

The results consist of execution time per step. and the outcome of physiological variables: heart rate, GSR, and EEG. Each will be discussed in turn.

### 4.1 Execution time

Before describing the analysis and results relating to execution time, it should be noted that participants' performance during the execution of the task was error free; all steps were completed as instructed.

Fig 6 shows the mean per-step execution time per block, averaged over block length (number of steps) and artificial delay. We can clearly see a trend for the execution time to increase with the delay, increasingly so for longer blocks. Since the per-step execution time compensates for the added delay in the measure *per se*, this trend must result from the participants' behavioral adjustment to the delay. This leads to one of the key outcomes from this study: *participants tend to act more slowly on steps affected by longer delays*—i.e. there is evidence of a pacing effect in users' behavior with respect to the responsiveness of the system.

We confirmed this effect through an *analysis of variance* (ANOVA [51]) with factors of block length and delay, and found significant main effects of both factors and the interaction, shown in Table 2.

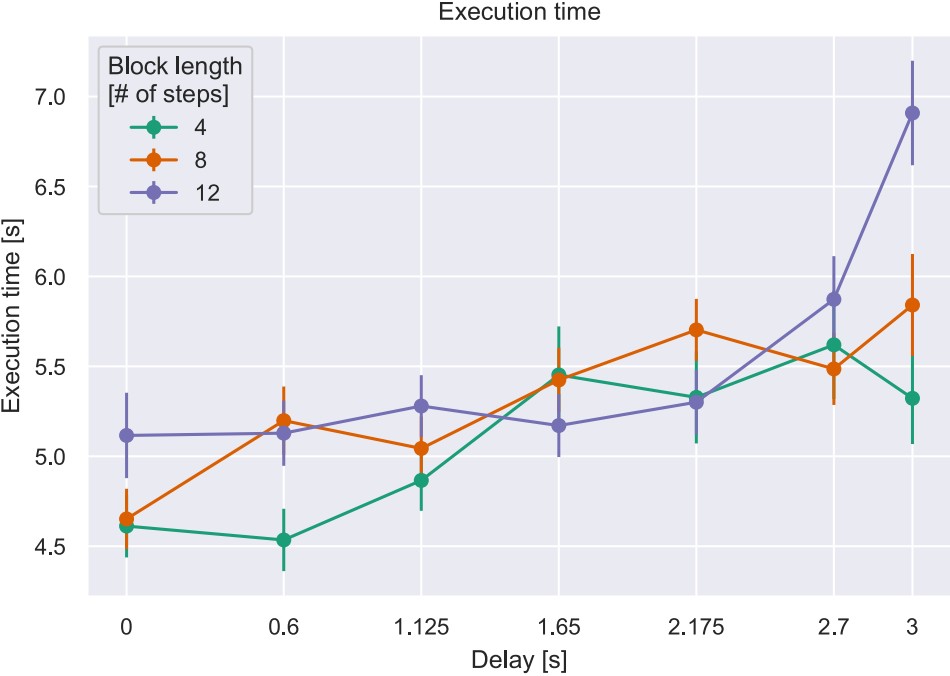

**Fig 6. Per-step execution time by block length vs. delay.** Error bars indicate the Standard Error of the Mean (S.E.M.).

Further analysis focused on the progressive effect on per-step execution time as additional steps occurred at a constant delay. For this purpose, the steps within a block were aggregated over sequences of 4, constituting a *slice*. Note that the first slice within a block is procedurally identical for all block lengths, in the sense that a participant currently performing a step in the first slice of a block has no way of predicting if the block ends after step 4 or not. The same logic can be applied to slice 2 for blocks of length 8 and 12. Accordingly for each participant, slice 1 data (steps 1–4) were pooled over all 3 block lengths, slice 2 (steps 5–8) were combined for block lengths 8 and 12, and slice 3 comprised the last four steps in blocks of length 12. An ANOVA on slice number (1–3) and delay (7 values) yielded effects of slice number, delay, and the interaction, detailed in Table 3. As shown in Fig 7, blocks with shorter delays showed a trend for the execution time to progressively decline over the course of the block (i.e., by slice

**Table 2. Significant effects on per-step execution time from ANOVA on factors delay and block length.**

| Factor | F-test | $p$ | $\eta_p^2$ |
|---|---|---|---|
| length | $F(2, 78) = 9.59$ | <0.001 | 0.20 |
| delay | $F(6, 234) = 15.52$ | <0.0001 | 0.28 |
| length × delay | $F(12, 468) = 3.84$ | <0.0001 | 0.09 |

**Table 3. Significant effects on per-step execution time from ANOVA on factors block slice and delay.**

| Factor | F-test | $p$ | $\eta_p^2$ |
|---|---|---|---|
| slice | $F(2, 78) = 88.79$ | <0.0001 | 0.69 |
| delay | $F(6, 234) = 14.13$ | <0.0001 | 0.27 |
| slice × delay | $F(12, 468) = 2.49$ | <0.01 | 0.06 |

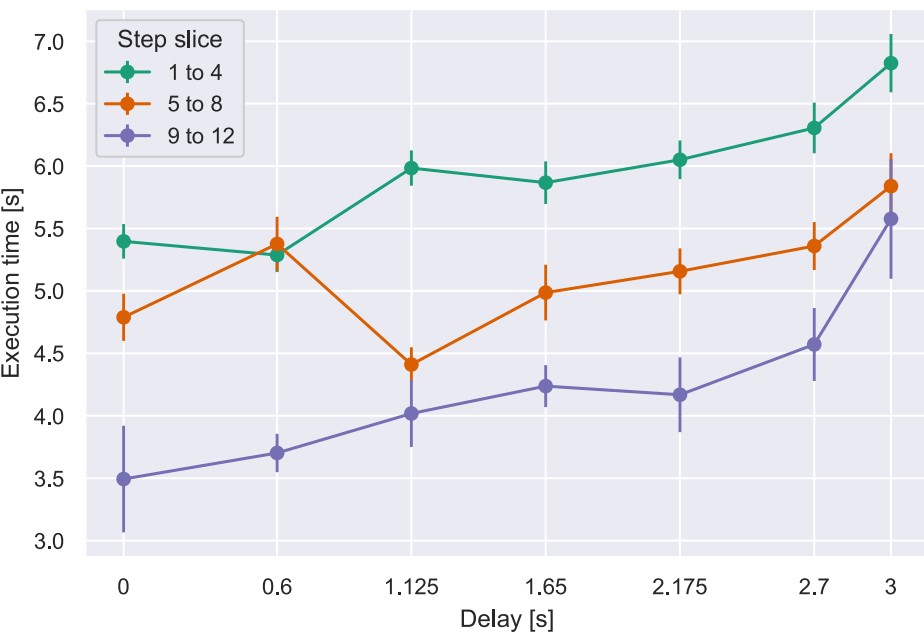

**Fig 7. Mean per-step execution time vs. delay, by step slice.** Error bars indicate S.E.M.

number), indicating that the participant accommodated to the feedback pace with more efficiently timed responses. With the longest delays, where the execution time per step was longest, the slow-down persisted; that is, the system response time hindered the participant's execution throughout the course of the block.

The data also allowed us to perform sub-analyses to assess the effects of carryover from one delay to another. Specifically, we measured the per-step execution time for the first four steps of a block when participants transferred from a relatively long delay (2.175—3.0 s) versus a short delay (0—1.65 s). Note that use of the first four steps controls for block length. We performed analyses where participants transitioned from either a short- or long-delay block into a: (i) no delay block (36 subjects) (ii) 1.65 s delay block (40 subjects) (iii) 2.7 s delay block (40 subjects) (iv) 3.0 s delay block (37 subjects). The destination delays were chosen so as to maximize the number of participants which contributed samples to the analyses, the results of which are pictured in Fig 8. We found that transitions from long-delay blocks carried over to significantly increase the per-step execution time of initial steps in the destination block for three of the four transitioned-into delays that were assessed (0, 2.7, 3.0 s, all $p < .025$).

In summary, we observe two direct effects on execution time due to added delays. Firstly, we see a clear hampering of the improvement of execution time across steps that is otherwise evident across blocks. Secondly, we notice that this effect lingers on even after the delay is removed, affecting subsequent blocks in the task.

## 4.2 Acceleration

Acceleration data from the E4 wristband were taken in 3 axes defined relative to the device. As our interest was in the amount of movement rather than the direction in space, we calculated a "movement score" for each block which we defined as:

$$M_B = \frac{\sum_{A_B} |\vec{\alpha_j}|}{\Delta T_B} \qquad (1)$$

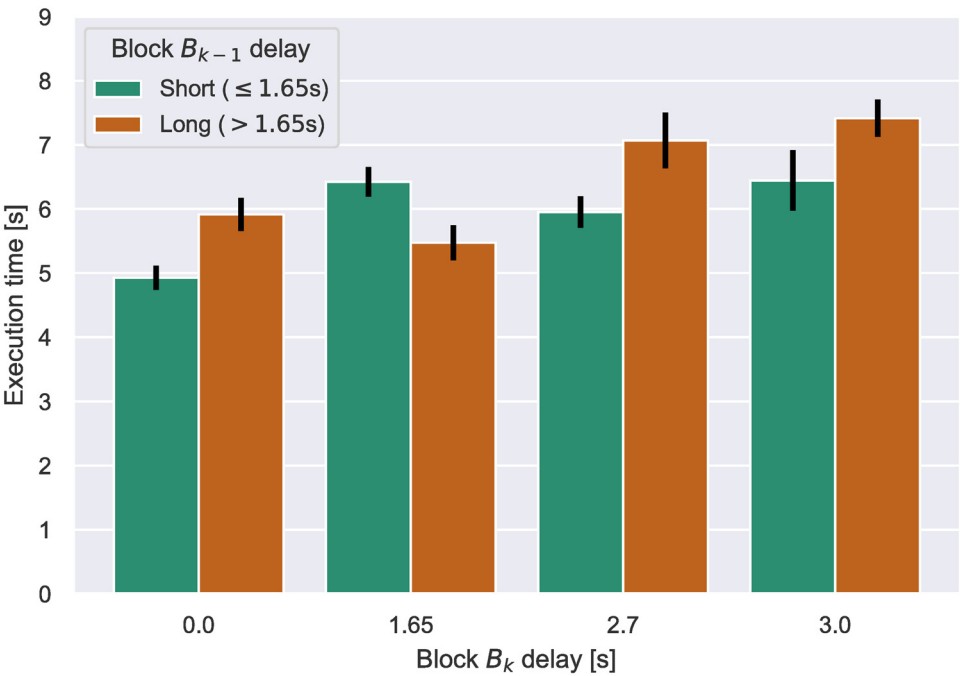

**Fig 8. Per-step execution time across the first four steps after a block transition from block $B_{k-1}$ to $B_k$.** Error bars indicate S.E.M.

where $B$ is an arbitrary block, and $A_B = \{\vec{\alpha}_0, \vec{\alpha}_1, \ldots, \vec{\alpha}_k\}$ represents the set of acceleration vector samples collected for said block. This score would include the time imposed by the delay and the time to decode the instructions and respond by moving the pieces so that the next state was recognized. It is only during the last part of the step that explicit movement is required, so any additional accelerations would derive from arbitrary movements while processing and waiting. We normalized the sum of the accelerations by the duration of the step to correct for differences in delay and the execution time, which tends to increase with delay as described above. An ANOVA on block length and delay showed a significant effect such that movement score decreased with delay, as shown in Table 4 and Fig 9. There was also a significant delay by length interaction, reflecting that delay effects occurred particularly for the longer lengths.

We next conducted the same analysis as for execution time, dividing blocks into three "slices" of four steps each, such that slice 1 comprised the first four steps of all block lengths, slice 2 the second four steps of blocks of length 8 and 12, and slice 3 the last four steps of blocks 12 steps long. As shown in Fig 10 the effect of delay was attributable only to the later slices, yielding main effect of slice number and delay and an interaction (Table 5).

These results present further evidence of the aforementioned pacing effect. As a sequence of steps with a long delay unfolds, more of the step duration is spent without moving. This can be interpreted in the context of the increased execution time at long delays demonstrated in

**Table 4. Significant effects on accelerometer data from ANOVA on factors delay and block length.**

| Factor | F-test | p | $\eta_p^2$ |
|---|---|---|---|
| delay | $F(6, 234) = 18.56$ | $<0.001$ | 0.32 |
| length × delay | $F(12, 468) = 3.04$ | $<0.001$ | 0.07 |

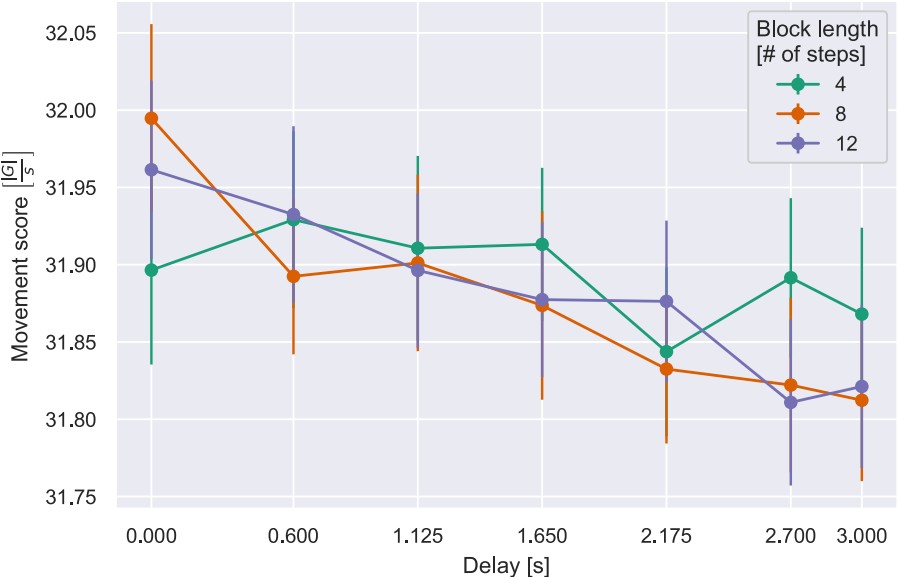

**Fig 9. Movement score vs. delay, per block length.** Error bars indicate S.E.M.

the previous analysis. Assuming that adding or deleting a LEGO block takes essentially the same amount of time and accelerates the wrist similarly at any one step, it appears that the participant simply remains stationary during the extra time that is induced by a series of long delays. Accordingly, the acceleration per unit time, our movement score, is reduced.

## 4.3 EEG

Analyses were conducted on the log EEG power in the alpha band, beta band, and total of all bands measured. Readings from the two frontally placed poles were highly correlated and were

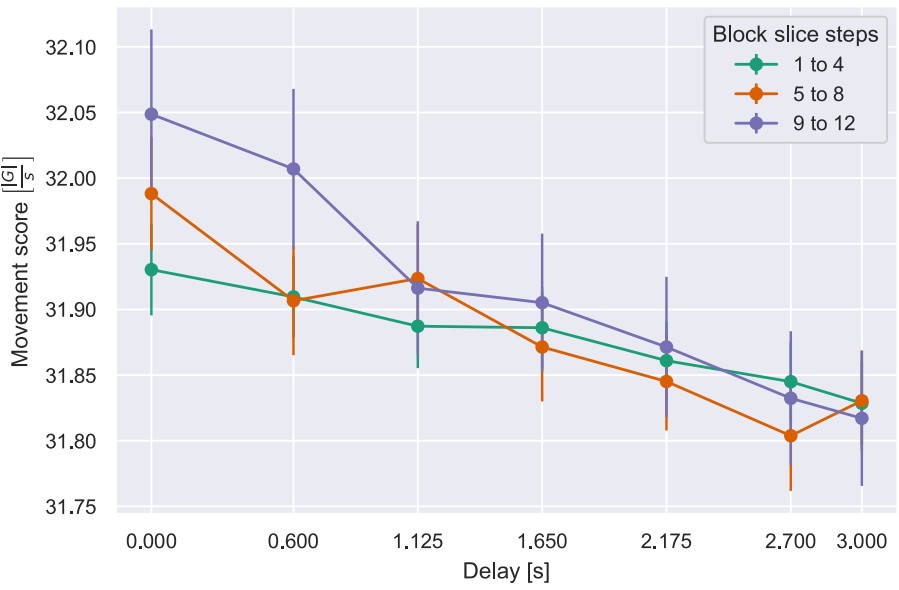

**Fig 10. Movement score vs. delay, per block slice.** Error bars indicate S.E.M.

**Table 5. Significant effects on accelerometer data from ANOVA on factors delay and slice number.**

| Factor | F-test | $p$ | $\eta_p^2$ |
|---|---|---|---|
| slice | $F(2, 78) = 8.39$ | <0.01 | 0.18 |
| delay | $F(6, 234) = 26.92$ | <0.001 | 0.41 |
| delay × slice | $F(12, 468) = 4.08$ | <0.001 | 0.1 |

**Table 6. Significant effects on log EEG power from ANOVA on factors delay and block slice.**

| Band | Factor | F-test | $p$ | $\eta_p^2$ |
|---|---|---|---|---|
| alpha | slice | $F(2, 78) = 28.26$ | <0.001 | 0.51 |
| beta | slice | $F(2, 78) = 29.18$ | <0.001 | 0.52 |

pooled into an average. Logs were taken for analysis because the EEG distribution tended to have a rightward tail. Twelve participants were excluded from the analysis, 9 due to device failure and 3 because of extreme values (i.e., the participant mean of log total power was greater than 3 s.d.s. from the mean of all participants).

The analyses then comprised 28 participants. Omnibus ANOVAs were conducted with delay and block length (number of steps) as factors, on the EEG data from the alpha and beta band. The analysis of alpha EEG found no significant effects. Beta EEG showed only an effect of block length, $F(2, 54) = 3.56$, $p = .035$, $\eta_p^2 = .12$, reflecting a tendency for the 4-step length to produce lower log power (mean 3.9 vs. 4.1 for lengths of 8 and 12). However, this effect was small and not consistent across delays.

Again the analysis dividing blocks into 4-step slices was conducted, with delay and slice number as factors. For both the alpha and beta bands, ANOVAs yielded effects only of slice number—see Table 6. Both bands showed the same tendency: EEG declined as a sequence of steps with the same delay progressed. These effects are shown in Fig 11. Thus, EEG taken from

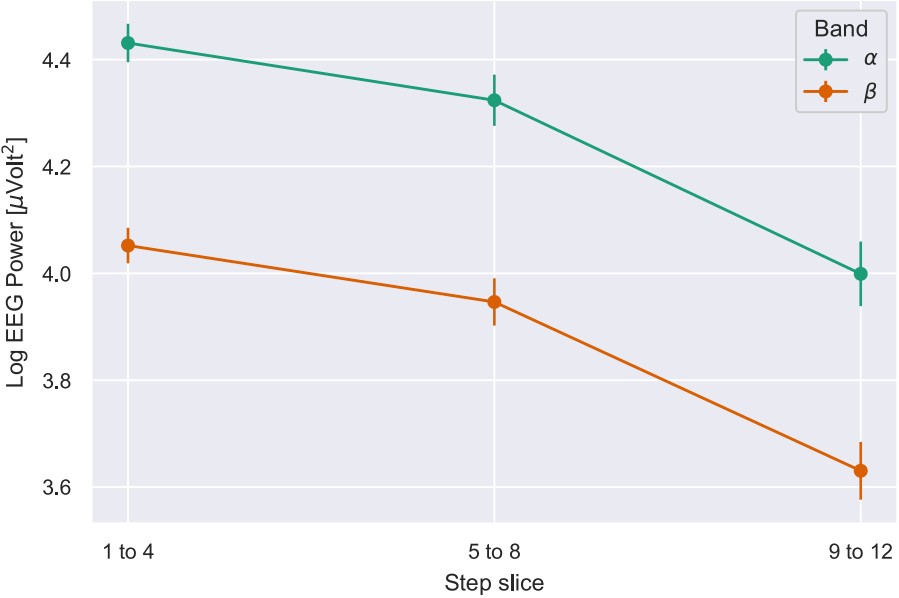

**Fig 11. Log of the average EEG Power for alpha and beta bands per step slice.** Error bars indicate S.E.M.

frontal locations mimics the execution time data in showing a decline over the course of a block, but unlike the execution time, there was no tendency for the decline in EEG to be reduced at longer delays.

## 4.4 Galvanic skin response (GSR) and heart rate

The measure of GSR was the log of total amplitude in the signal. To control for effects of block length (steps × delay) and the slow-down in execution time at longer delays, GSR data were normalized by the temporal duration of the block. Due to sensor failure and the elimination of one subject with extreme values of GSR amplitude using the same rule as for EEG, the analysis comprised 34 participants. No systematic trend due to delay or block length was observed. In addition, the same analysis of slice position within block length as was performed for the EEG revealed no effects, indicating that GSR was stable across step positions within a block.

Heart rate, measured in beats per minute, averaged 90.2 BPM and showed no trend related to the experimental variables of delay and block length. The same was true of the variability in beats per minute (average standard deviation across conditions was 20.3).

The absence of systematic trends in both these results is interesting in the context of our initial suggestions of potential mechanisms relating delay in the application to human behavior. In section 2.2 we proposed that emotional arousal during the execution of the task was a potential explanation for the effects observed in the human. However, these results seem to refute this hypothesis, and will be further discussed in Section 5.

## 4.5 Individual difference analysis

The Big 5 personality inventory and Immersive Tendencies Questionnaires (ITQ) were combined with outcome variables in an analysis of individual differences. Given the previous results, we initially considered the following outcome variables:

- execution time in the most demanding block (length 12, delay 3.0 s);

- total of alpha and beta bands EEG in the first four steps at all lengths ("Slice 1");

- average heart rate;

- and average log GSR.

Although the heart rate and GSR data had produced no significant effects in the analysis of experimental variables, they could in principle correlate with experimental outcomes across individuals. A principal-components analysis (*PCA*) on a subset of these variables, shown in Table 7, was ultimately conducted on the 28 participants for which there was EEG data. Three of the personality measures were excluded after initial analyses indicated significant correlations among neuroticism, openness, agreeableness, and extroversion. Neuroticism (with poles of sensitivity and security) was selected as most relevant to the issue of response to system delay and was included along with conscientiousness. GSR was used in a subsidiary analysis, because only 25 participants had both EEG and GSR measures that were reliable.

The PCA produced three components that accounted for 73.13% of the variance in the six factors considered. Component 1 included neuroticism, both ITQ scores, and execution time, indicating that more sensitive and immersed individuals tended to slow their responses under extended delay. Component 2 included conscientiousness and the absence of neuroticism along with immersive involvement, indicating that efficient and secure participants tended to be more involved. The third component had positive loadings on EEG in Slice 1 (first four steps of a block) and the focus component of the ITQ. An additional analysis including heart rate added a component but improved the PCA fit by only 4.5%, indicating that any variability

**Table 7. Principal component analysis.**

**(a)** Main components identified.

| Factor | Comp. 1 | Comp. 2 | Comp. 3 |
|---|---|---|---|
| BFI Conscientiousness | −0.022 | 0.668 | −0.481 |
| BFI Neuroticism | 0.600 | −0.678 | −0.118 |
| ITQ Focus | 0.678 | 0.203 | 0.504 |
| ITQ Involvement | 0.573 | 0.540 | 0.417 |
| Exec. Time (delay 3.0 s, length 12) | 0.758 | −0.178 | −0.348 |
| Log EEG power $\alpha + \beta$ Slice 1 | −0.436 | −0.251 | 0.589 |

**(b)** Percentages of variance explained by the components.

| | Comp. 1 | Comp. 2 | Comp. 3 | Total |
|---|---|---|---|---|
| Explained Variance | 31.88% | 22.22% | 19.03% | 73.13% |

in this measure across individuals is unrelated to personality, immersiveness, or outcome. A further analysis including GSR produced a solution in which GSR loaded with the first component, along with execution time.

Fig 12 shows the correlation between neuroticism and execution time for the block with extreme values of length (12 steps) and delay (3.0 s), for the full set of 40 participants. It confirms the relationship indicated by the first component in the PCA using the smaller sample of 28 participants; that is, higher neuroticism is associated with more responsiveness to delay.

On the whole, these results suggest that individual differences in widely accepted personality variables and immersive tendencies moderate the response to delay. This fact could have practical implications in the future. It could, for instance, provide a tunable parameter for eventual models aiming to emulate human interaction with a WCA. In addition, physiological

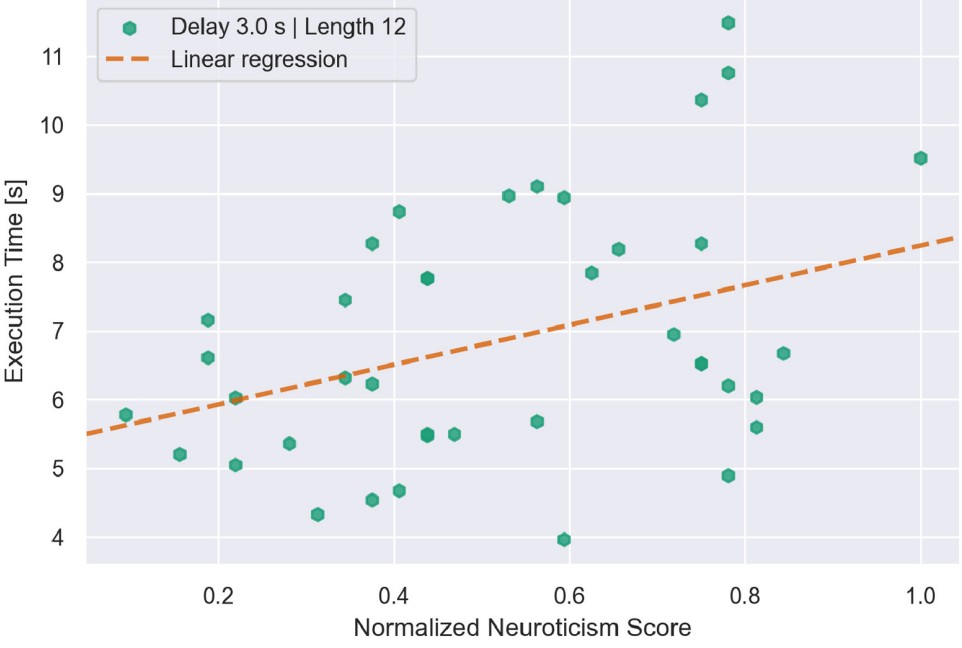

**Fig 12. Correlation between neuroticism score of participants and their execution time in the longest block at the longest delay.** Pearson correlation coefficient $r = .40$; 2-tailed $p = .01$.

measures of heart rate and EEG appear not to be direct indicators of behavioral response to delay, although GSR may be more promising in this regard.

## 5 Discussion

We start our discussion with the main results of our experimentation presented in the previous section:

- Firstly, and perhaps most importantly, we find that a system slow-down induces an *additional* behavioral slow-down. That is, as system responsiveness decreases, our data indicates that users significantly slow-down in their execution of the task. This slow-down scales with the decrease in responsiveness; compared to the no-delay case, participants were on average 12% slower at 1.65 s delay and 26% at 3.0 s delay. Moreover, there is a temporal component to this effect; users become progressively slower the more time passes with reduced system responsiveness.

- Secondly, we find that the effects of behavioral slow-down due to impaired system responsiveness *remain* for at least a few steps after system responsiveness improves. This is evidenced by the longer per-step-execution times of the first four steps of blocks immediately following a high-delay block, as pictured in Fig 8. The question of whether any lingering effect can be measured after these four steps remains open.

- Thirdly, we evidence a speed-up in execution time over a series of steps; that is, subjects get faster at performing steps as the task progresses. However, the strength of this effect decreases as delay increases. Whereas for blocks without delay users performed the last four steps of a 12-step block on average 36% faster than the first four, at the maximum delay this effect practically disappears.

- Fourthly, in terms of inter-subject differences, PCA revealed three main factors governing users' response to delay. The first factor represents sensitivity to delay as moderated by the "Big Five" personality trait of neuroticism and both measures of immersion: focus and involvement. Factor two and three represent dedication to the task as opposed to delay intolerance and reflect variables related to attentiveness, respectively. In simple terms, these results suggest that the effects of delay are most potent in individuals who are sensitive and involved in the task. The findings appear selective to cognitive assistance tasks like the present ones, inasmuch as the same measures did not correlate with outcomes in other computer-intensive environments such as immersive VR [52]. These correlations are also consistent with previous findings indicating that individuals scoring high in neuroticism tend to be intolerant to delayed reward [13].

A central question therefore arises: to which physio- and psychological mechanisms can these findings, most importantly the substantial slow-down in task execution, be attributed?

In Section 2, we initially considered the possibility that delays might produce negative emotional reactions. These could in turn elicit generalized arousal. We also postulated that adapting to delay might progressively deplete cognitive resources in users. However, the present data provide relatively little support for these alternatives, in that the predicted measures did not produce the expected statistical trends. Specifically, physiological measures of GSR and HR failed to show evidence of differential arousal under long vs. short delays, and speed-induced errors and non-completions predicted by resource depletion were not observed. The acceleration data further did not indicate that extended delay significantly increases erratic movement. To the contrary, the data suggest this effect results from a delay in movement after an instruction is introduced. That is, users fail to capitalize on the new information as quickly

as they could. Thus, contrary to our preliminary postulations, behavioral effects seem to arise from impaired cognitive control mechanisms, and not from emotion or resource depletion. We hypothesize that the effects of feedback latency can best be understood as changes in the use of a cognitive plan. As was described in Section 3 and Fig 3, complex cognitive and motor tasks have been modeled as the unfolding of a hierarchy of command, from high-level plans to physical output. Long system latencies, we propose, disrupt the automating of such a plan, instead relegating it to attention-based control at the step-by-step level that is easily diverted. This also provides a possible explanation for the lingering effects of delay after an acceptable system responsiveness is restored, as users needs time re-adjust and re-automate their cognitive plans.

As to the applicability of our findings to other applications, it must be noted that these results pertain to a specific class of applications, namely step-based task-guidance WCA. However, we would expect our findings to extend to similar applications, as long as they follow the same pattern of seamless interaction—i.e. such that the user does not need explicitly interact with the application to advance the state.

The results here presented provide a number of possible implications for WCA system design and optimization, both for single and multi-application flows.

- Due to the behavioral slow-down in users, even short-term reductions in responsiveness will lead to significantly extended application lifetimes. This has direct implications for resource and power consumption.

- The fact that the adverse effects of delay on users do not immediately disappear as the system returns to a high-responsive state could have unconventional consequences for resource allocation. This is of particular importance, for instance, for cases where the user may be able to finish the task before these effects subside. In such cases, the limited potential gains might not justify diverting valuable system resources to the impaired application.

- In multi-user environments, the time dependency of user slow-down effects mean that fair degradation of system responsiveness across applications may not ultimately be beneficial to the system as a whole. Take for instance two applications on the same system which negatively interfere with each other. The longer they interfere with each other, the longer their respective lifetimes are going to be, which in turn causes them to interfere even longer, potentially entering a positive feedback loop. In such a case, prioritizing one over the other rather than trying to improve responsiveness for both might lead to resources being freed up faster system-wide.

- Based on our findings relating individual differences between users and their sensitivity to delays, it might also be possible to extrapolate user characteristics from measured execution times. This could prove a valuable tool for load balancing, for instance by prioritizing resource allocation to users with a higher sensitivity to system-state degradation. However, this remains an open challenge.

To wrap up, we believe the present data provide novel and unexpected insights for the understanding and optimizing of WCA deployments. Although more subtle than expected, and in some cases somewhat counterintuitive, these insights represent a valuable tool to tackle inefficiencies in these systems. Moreover, we also argue these findings represent a first step towards a full-fledged understanding of the relationship between application responsiveness and human behavior. More research in this area will surely uncover more complex and interesting behaviors. Finally, we believe the data provide parameters that can usefully be integrated into cognitive models of WCA that might be constructed under existing architectures like

ACT-R. These same parameters could be used to modulate the timing and generation of inputs in trace-based workload generation tools such as the EdgeDroid platform [15, 16], allowing the tool to use the same trace to generate workloads for a multitude of different user profiles.

## 6 Conclusion

In this paper, we presented the results of a study on the physiological and behavioral reactions of users of WCA to delays in the application pipeline. Our ultimate aim was to identify and categorize the ways in which humans react to low system responsiveness in step-based cognitive assistance systems.

We approached this in an experimental manner, by having participants interact with an instrumented WCA setup, and found that delay appears to affect the cognitive plan of users, preventing them from automating the task they are performing. The results show that user interactions in a WCA slow down in the presence of delays in the application pipeline. When system responsiveness is high, the user responds quickly; when it is low, the user slows down. This was evidenced by an increase in task execution times, even after accounting for the artificially introduced delays, as well as accelerometer data from participants wrists. Additionally, we found that the strength of this effect is modulated by individual differences between subjects.

These results are interesting as they open up hitherto unexplored opportunities for the design, optimization and benchmarking of WCA systems. In this context, we believe there are two direct and important next steps to be performed. The first of these relates to the implications for system optimization and resource allocation discussed in Section 5. We believe these need to be implemented, tested and validated in real setups. In particular, we identify two of these implications to be prime candidates for their own experimental studies. One, our postulation that in cases where an impaired application is close to finishing, diverting resources to it might not be the most optimal course of action; and two, the possibility that "fair" degradation of system responsiveness across applications may be, in some cases, undesirable. Both of these questions could be answered with straightforward setups.

The second step corresponds to the extension of existing tools for WCA benchmarking with the findings presented in this paper. These tools are of great usefulness for the study of WCA systems, as they allow for automated large-scale testing without having to resort to human users. However, they are still somewhat simplistic and unrealistic in their workload generation schemes. Incorporating the results presented here would allow for much more realistic workloads. For instance, our findings relating to the effects of delay on execution times could be directly adapted to modulate timings in the input stream. Another example would be employing the results linking neuroticism to a heightened sensibility for delays to provide a "tuning knob" for the user models in these workload generation schemes. Extending and perfecting these tools will allow for much more realistic benchmarking and testing of WCA systems, providing data of significantly better quality and ultimately leading to faster improvement, optimization and adoption of these systems.

## Acknowledgments

Any opinions, findings, conclusions or recommendations expressed in this material are those of the authors and do not necessarily reflect the view(s) of their employers or funding sources.

## Author Contributions

**Conceptualization:** Manuel Olguín Muñoz, Roberta Klatzky, Mahadev Satyanarayanan, James Gross.

**Funding acquisition:** Mahadev Satyanarayanan, James Gross.

**Investigation:** Manuel Olguín Muñoz, Roberta Klatzky, Junjue Wang.

**Methodology:** Manuel Olguín Muñoz, Roberta Klatzky, Junjue Wang, Padmanabhan Pillai, James Gross.

**Resources:** Roberta Klatzky.

**Software:** Manuel Olguín Muñoz.

**Supervision:** Roberta Klatzky, Mahadev Satyanarayanan, James Gross.

**Validation:** Roberta Klatzky, Padmanabhan Pillai, James Gross.

**Visualization:** Manuel Olguín Muñoz.

**Writing – original draft:** Manuel Olguín Muñoz.

**Writing – review & editing:** Manuel Olguín Muñoz, Roberta Klatzky, Junjue Wang, Padmanabhan Pillai, Mahadev Satyanarayanan, James Gross.

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
