## [Decision Letter · Decision Letter 0]

6 Jan 2021

PONE-D-20-32471

Impact of delayed response on Wearable Cognitive Assistance

PLOS ONE

Dear Dr. Olguín Muñoz,

Thank you for submitting your manuscript to PLOS ONE. After careful consideration, we feel that it has merit but does not fully meet PLOS ONE’s publication criteria as it currently stands. Therefore, we invite you to submit a revised version of the manuscript that addresses the points raised during the review process.

We look forward to receiving your revised manuscript.

Kind regards,

M. Usman Ashraf, Ph.D

Academic Editor

PLOS ONE

Journal Requirements:

2.) We note that you have indicated that data from this study are available upon request. PLOS only allows data to be available upon request if there are legal or ethical restrictions on sharing data publicly. For more information on unacceptable data access restrictions, please see http://journals.plos.org/plosone/s/data-availability#loc-unacceptable-data-access-restrictions.

Reviewers' comments:

Reviewer's Responses to Questions

**Comments to the Author**

1. Is the manuscript technically sound, and do the data support the conclusions?

Reviewer #1: Yes

Reviewer #2: Yes

2. Has the statistical analysis been performed appropriately and rigorously? 

Reviewer #1: I Don't Know

Reviewer #2: Yes

3. Have the authors made all data underlying the findings in their manuscript fully available?

Reviewer #1: No

Reviewer #2: No

4. Is the manuscript presented in an intelligible fashion and written in standard English?

Reviewer #1: Yes

Reviewer #2: Yes

5. Review Comments to the Author

Reviewer #1: Note.

Following page numbers refer to the numbers at the bottom of pages which are in black colour in the manuscript.

Line numbers refer to the line numbers of the document that are written on the right side of the pages.

I have several comments regarding different parts of the manuscript that follow below:

Introduction

1)Please bring references for “One is providing quality of life improvements to the millions of people around the world affected by some form of cognitive decline”

“WCA can, for instance, provide assistance to people recovering from traumatic brain injuries, smoothly guiding them through day-to-day interactions with the world which would otherwise be extremely challenging”.

(p2, lines 13-17).

2) “Characterizing the relationships between system responsiveness and user behavior and experience is of paramount importance for the design and evaluation of these applications.” (p2, lines 54-56).

This sentence needs some more clarifications regarding user behaviour and experience:

Bringing some examples for user behaviour and specifying the experience may help to better understand it. For instance, if the experience is an experience about the use of such applications then it needs to add “use experience” to make it clear.

3)The research questions cannot be found easily. They should be explained. The aim of the study is mentioned (p3, lines 64-65) i.e. how human behavior changes with system responsiveness. We aim to tackle this question…

but the research questions need to be clarified too.

4)We present in this paper an experimental WCA test-bed of our design and making.

(p3, line 70) it seems the sentence is not complete: and making??

5)This test-bed was subsequently employed in a study in which undergraduate students were asked to interact with and follow the instructions given to them by a cognitive assistant (p3, lines 71-73).

Please bring more information about undergraduate students for example in which field, gender…

6)Research hypotheses should be mentioned and explained in a clear way.

7)The introduction part needs to be more elaborated with theoretical frame.

Section 2.2

1)As the authors wrote a third potential explanation of delay effects…(p5,line 167) 

Similar types of writing for all three explanation may help readers grasp the messages easier:

Thus, I suggest that the authors bring at the beginning of previous relevant phrases “first explanation” and “second explanation” too.

Section 3: experimental design

1)Key characteristics of the sample should be clarified.

2)Sections 3.1, 3.2 need to be summarized. This part regarding definitions are too long and need to be shortened.

3)Subsections 3.3.3 also need to be shortened and summarized in a clearer way. This part is very long too. For instance, EEG is well known, it doesn’t need to be defined very much.

Discussion

1)This part is very short in comparison to other parts of this study. The reflection of the authors of the current paper on their results is missing (the reflection of authors regarding why some results were significant while other not are important). Thus the authors should elaborate the discussion part as well.

Reviewer #2: The paper is well written. Considering the following comments makes the manuscript better:

1) The explanations have been used too much in somewhere. For example, it is not necessary to explain the ANOVA test. This test is well known.

2) Use more resolution for figures.

3) Some figures can be inserted in a figure as subplots and or subpanels.

4) In line 286 at the end of line the "A Latin" is written, what is that??

6. PLOS authors have the option to publish the peer review history of their article (what does this mean?). If published, this will include your full peer review and any attached files.

Reviewer #1: No

Reviewer #2: No

---

## [Author Response · Author response to Decision Letter 0]

17 Feb 2021

PONE-D-20-32471

Impact of delayed response on Wearable Cognitive Assistance

PLOS ONE

Dear Dr. Usman Ashraf:

We would like to begin by thanking you for giving us the opportunity to submit a revised version of our manuscript “Impact of delayed response on Wearable Cognitive Assistance”. We also wish to thank you and the reviewers for the effort you have dedicated to providing us with relevant and insightful comments to improve the manuscript. 

After careful consideration, most of these recommendations have been included and highlighted in the revised paper. In particular, we have reworked extensive segments of the Introduction — following Reviewer 1’s comments — as well as significantly summarized subsections 3.1, 3.2 and 3.3.3 — as per the combined suggestions of both reviewers. Please see the attached "Response to Reviewers" document for a detailed, point-by-point response to each of the comments and suggestions made by the reviewers.

With regards to the dataset associated with this research, we have chosen to make it publicly available in full, as a repository on the Zenodo platform:

• DOI: 10.5281/zenodo.4494912

• URL: https://zenodo.org/record/4494912

Once again, we thank the reviewers for their comments and hope we have managed to sufficiently address their concerns in this revision.

Sincerely,

Manuel Olguín Muñoz

First Author/Corresponding Author PONE-D-20-32471

Division of Information Science and Engineering, EECS

KTH Royal Institute of Technology

Malvinas väg 10, 7th floor

100 44 Stockholm, Sweden

molguin@kth.se

---

## [Editor Report · Decision Letter 1]

4 Mar 2021

Impact of delayed response on Wearable Cognitive Assistance

PONE-D-20-32471R1

Dear Dr. Olguín Muñoz,

We’re pleased to inform you that your manuscript has been judged scientifically suitable for publication and will be formally accepted for publication once it meets all outstanding technical requirements.

Kind regards,

M. Usman Ashraf, Ph.D

Academic Editor

PLOS ONE

---

## [Editor Report · Acceptance letter]

8 Mar 2021

PONE-D-20-32471R1 

Impact of delayed response on Wearable Cognitive Assistance 

Dear Dr. Olguín Muñoz:

I'm pleased to inform you that your manuscript has been deemed suitable for publication in PLOS ONE. Congratulations! Your manuscript is now with our production department. 

Kind regards, 

on behalf of

Dr. M. Usman Ashraf 

Academic Editor

PLOS ONE